# Fluorescent In Situ Hybridization for the Detection of Intracellular Bacteria in Companion Animals

**DOI:** 10.3390/vetsci11010052

**Published:** 2024-01-22

**Authors:** Matthew J. Rolph, Pompei Bolfa, Sarah M. Cavanaugh, Kerry E. Rolph

**Affiliations:** Center for Integrative Mammalian Research, Ross University School of Veterinary Medicine, Basseterre P.O. Box 334, Saint Kitts and Nevis

**Keywords:** intracellular bacteria, fluorescent in situ hybridization, inflammatory disease, dogs, cats

## Abstract

**Simple Summary:**

Bacterial infections have traditionally been identified by culture; however, with advances in the understanding of the role of bacteria in disease, it is recognized that intracellular bacteria can be visualized within cells. Fluorescent in situ hybridization (FISH) is a technique used to not only identify intracellular organisms but also allows researchers to see them, in situ. In this review, we discuss the use of FISH for the identification of intracellular bacteria in companion animals. Most studies have focused on the identification of *Escherichia coli*, particularly within the gastrointestinal tract. Additionally, bacterial associations with inflammatory diseases affecting the liver, kidney, skin, and cardiovascular systems have been investigated with mixed results.

**Abstract:**

FISH techniques have been applied for the visualization and identification of intracellular bacteria in companion animal species. Most frequently, these techniques have focused on the identification of adhesive-invasive *Escherichia coli* in gastrointestinal disease, although various other organisms have been identified in inflammatory or neoplastic gastrointestinal disease. Previous studies have investigated a potential role of *Helicobacter* spp. in inflammatory gastrointestinal and hepatic conditions. Other studies evaluating the role of infectious organisms in hepatopathies have received some attention with mixed results. FISH techniques using both eubacterial and species-specific probes have been applied in inflammatory cardiovascular, urinary, and cutaneous diseases to screen for intracellular bacteria. This review summarizes the results of these studies.

## 1. Introduction

The origins of fluorescent in situ hybridization (FISH) can be traced back to 1969 when in situ hybridization (ISH) staining techniques using radioactive isotopes as markers were developed by Gal and Pardue [1]. These were used to identify specific nucleic acid sequences within tissues by binding a nucleotide probe labelled with a reporter molecule to the desired region for identification [2,3,4]. 

The first reported use of fluorescent dyes to enhance in situ hybridization was documented by Bauman, et al. [5], in which a 3′-fluorescence-labeled RNA probe was used to bind to a specific DNA sequence. However, at its inception, FISH techniques demonstrated non-specific binding due to large probe size (which often contained repeat DNA sequences) and led to excessive background fluorescence, rendering results questionable [6,7]. Improvements to FISH in the form of reduced probe length, probe type, the fluorochromes/fluorophores used, and increased mechanization have helped negate the issues experienced in its formative stages [2,7,8,9,10,11,12]. These advances have led to FISH being used in neuroscience, reproductive medicine, toxicology, evolutionary biology, and microbial ecology. FISH is regularly used in histopathological studies, as well as being considered the ‘gold standard’ in cytology and genomics for the detection of genetic aberrations resulting in diseases such as non-small cell lung cancer, breast cancer, urothelial carcinomas, and numerous forms of leukemia, to name but a few [13,14,15,16,17,18,19,20]. 

The successful application of fluorescence in situ hybridization is reliant on a number of fundamental steps, including specimen fixation (specifically, the fixation medium and duration of fixation); sample preparation, including any pre-hybridization steps that maybe required (e.g., permeabilization of cellular membranes); the addition and hybridization of probes to the desired target DNA sequence; the removal of unbound probes by washing; and the visualization of results using a fluorescence microscope and camera to document findings [10]. Furthermore, with respect to probe selection, there are a number of factors that need to be considered: the desired sequence to be targeted and thus the probe sequence to be used; the length of the sequence to be used; how that sequence is to be obtained (for example, through nick translation or with the use of synthetic oligonucleotides); whether to use deoxyribonucleic acid (DNA), ribonucleic acid (RNA), peptide nucleic acid (PNA), or locked nucleic acid (LNA) probes; the type of fluorophore to use; and whether there is a requirement for multiple probe/fluorophore combinations [21]. It is important to bear in mind that these factors may vary depending on sample type and the type of FISH being undertaken. For example, in contrast to DNA or RNA probes, which are more commonly used to investigate certain types of bacteria, PNA probes (used in PNA-FISH) may allow for the permeabilization steps to be omitted, as the probes readily penetrate cell membranes due to their relative lack of ionic charge and hydrophobic nature [22,23,24].

As a process, FISH is not without its challenges; for example, some techniques are not standardized, with significant variability arising from which disease is being investigated, the type of tissue involved, and how the sample has been prepared and/or stored. It also requires skilled and experienced individuals to perform the tests and interpret the results [25]. When investigating specific microorganisms, targeted investigations must precede FISH to identify the organism of interest. Probes can be tailored to the detection of a particular DNA/RNA segment, enhancing diagnostic sensitivity and specificity [11]. However, creating new probes for otherwise unexplored targets requires careful design and evaluation, with the need for rigorous checks and confirmation prior to use [26]. Even when Eubacterial probes are being used to screen for intracellular bacteria, it is important to be cognizant that certain bacterial walls may not allow penetration without the inclusion of enzymatic pre-digestion [27,28,29]. Where tissue samples are used, preparation times may be limiting and permeabilization steps need to be adjusted to account for sample type, risking insufficient permeability (resulting in insufficient probe penetration) or the destruction of cellular structures. Autofluorescence and nonspecific binding—if not sufficiently countered—may cause misleading results [30,31]. In addition, when analyzing primary materials (e.g., samples directly obtained from fresh tissues), FISH alone may not be as sensitive as PCR [2,7,10,11,32]. 

Regardless of these potential pitfalls, FISH is still regarded as a definitive process for the exploration/visualization of the spatial and morphological dynamics of chromosomes, DNA/RNA and other cellular structures, and the distribution of intracellular and extracellular organisms and bacterial consortia within a microenvironmental context [8,10,15,33,34]. Visualization also allows the quantification of the organisms present as well as whether the organisms were living or dead at the point of fixation (specifically, via the detection of the quantity of ribosomal RNA (rRNA) present within the organism) [11,35,36,37]. 

## 2. The Use of FISH for the Detection of Intracellular Bacteria in Dogs and Cats

When considering investigations into bacterial causes of disease within companion animals, especially where the pathogen may reside intracellularly, fluorescent in situ hybridization (FISH) has contributed significantly [11]. This technique is often used as a confirmatory test employed in conjunction with cell culture, histopathology, and PCR [38,39,40]. In this section, we present studies where a single, distinct bacterial species was identified as the primary cause of chronic disease using FISH as either the primary investigative method or in conjunction with others.

### 2.1. The Use of FISH in the Detection and Visualization of Escherichia coli

#### 2.1.1. *E. coli*-Induced Canine Granulomatous Colitis

To date, FISH has been used extensively in the investigation of causative agents in canine granulomatous colitis, helping to establish the presence of adherent strains of *Escherichia coli* capable of breaching the mucosa, as an organism of significant interest [41,42]. Regarded as the landmark paper for the detection of intracellular *E. coli*, Simpson et al. [43] first described the discovery of adherent-invasive *E. coli* and its association with the development of granulomatous colitis in Boxer dogs (GCB). Using FISH and eubacterial 16s rRNA probes alongside culture and 16s ribosomal DNA sequencing, it was found that *E. coli* were present in 100% of the CGB samples tested compared to none in the controls. Of the forms of *E. coli* discovered, three of four associated isolates were seen to be adherent -invasive and had replicated within cultured epithelial cells.

Mansfield et al. [44] corroborated these findings in their study, in which they used enrofloxacin to treat Boxer dogs with histiocytic ulcerative colitis (HUC). Using colonic biopsies taken from cases before and after treatment with enrofloxacin, it was found that intramucosal *E. coli* were eradicated in 4/5 cases, where HUC was in remission, supporting the causal relationship between *E. coli* and HUC. FISH was used with both eubacterial and *E. coli*-specific probes as the primary testing method, not only revealing the primary cause but also a resistant strain of *E. coli* in the one case that did not maintain a remissive state. Despite the apparent high sensitivity and specificity of FISH in this study, the number of cases included was low. 

A few case reports have also documented the use of FISH in the detection of canine *E. coli* infection. Conrado et al. [45] reported the use of FISH specific to the detection of *E. coli* to confirm cytology performed on rectal scrapings. The samples were collected from a French Bulldog diagnosed with granulomatous colitis, which was otherwise unresponsive to treatment. Despite initial treatments with enrofloxacin and enrofloxacin with cefpodoxime—both of which induced a remissive state—once treatment was stopped, clinical signs returned. The dog remained on a continuous course of enrofloxacin. Whilst FISH was used to confirm the cytological findings and aid diagnosis, the authors were unable to take additional tissue biopsies for subsequent analysis with FISH after treatment commenced, nor were they able to perform further cultures due to financial constraints. By their own admission, had they been able to do so, this may have aided in determining whether the intramucosal *E. coli* had been completely eradicated as well as establish further, more effective treatment regimes.

In a case series reported by Cochran et al. (2021) [46], five cases of canine granulomatous ileocolitis (GIC) were selected by a retrospective review of medical records between 2010 and 2014. These were positive for invasive mucosal *E. coli* of the ileum and colon based on periodic acid Schiff staining (PAS+). The report reviewed the diagnosis and treatment of the disease in four Boxer dogs and one French Bulldog from a total of 86 dogs with suspected granulomatous colitis (CG). The authors employed FISH using eubacterial and *E. coli*-specific probes on formalin-fixed paraffin-embedded (FFPE) samples to select the most appropriate cases. Intestinal biopsies were then taken and submitted for culture. Of the five cases included, all were treated with fluoroquinolones; all demonstrated a complete or partial response to treatment after 30 days. A complete response was noted to be prolonged in three cases (median interval = 40 months, 16–60), with two relapsing. These were subjected to repeat biopsies, which revealed multidrug-resistant forms of invasive *E. coli* within the mucosa of the ileum (1/2) and colon (2/2). This allowed for targeted therapies to be undertaken, which in turn led to a lasting partial recovery (78 months). This case series demonstrated the advantage of repeat biopsy in cases where treatment was unsuccessful, as well as noting that effective treatment was reliant upon testing bacterial susceptibility to guide antimicrobial therapy, which included the use of beta-lactam antibiotics.

In their case report describing the cytologic characteristics of canine granulomatous colitis, Sims et al. [47] used FISH to help identify and confirm the presence of *E. coli* in a male mixed-breed dog presenting with chronic bloody diarrhea and weight loss. Initially, rectal scrapings were obtained and submitted for cytological testing using Wright–Giemsa stain; mucosal samples were also obtained and submitted for culture and histologic examination using Hematoxylin and Eosin and Periodic-acid Schiff stains. All of the tests performed, including FISH, demonstrated the presence of rod-like bacteria consistent with *E. coli* within macrophages in the lamina propria. The patient was treated with a six-week course of enrofloxacin and a growth-formula diet. Subsequent follow-up examination at approximately nine months revealed weight gain and no recurrence of clinical signs.

#### 2.1.2. *E. coli* in Canine Focal Lipogranulomatous Lymphangitis

Whilst FISH has proven successful in the detection and/or confirmation of *E. coli* in many cases of granulomatous ulcerative colitis, Lecoindre et al. [48] found the technique to be of limited use in their study of focal lipogranulomatous lymphangitis (FLL). FISH, along with histopathology, was employed to evaluate bacterial and fungal cultures of samples taken from ten dogs. All had been selected by retrospective review of medical records demonstrating a histopathological diagnosis of FLL. On ultrasonography, all dogs showed signs of a thickened ileum/ileocolic junction and, as such, stenosis within the ileocolic opening prevented endoscopic intubation. Histology of resected samples from these regions showed evidence of granulomatous inflammation within the muscularis and serosa. FISH was also performed on these samples using 16s rRNA eubacterial probes and *E. coli*-specific probes, revealing the presence of invasive *E. coli* in only two of the ten dogs from which samples were obtained. Furthermore, the authors noted that the lack of response to PAS, Ziehl–Neelsen, and silver histochemical staining (also performed on the samples to detect mycobacterium, fungi, and prototheca) indicated that there was no infectious cause, potentially differentiating FLL from other forms of granulomatous ulcerative colitis as reported in Boxer dogs and French Bulldogs. However, the authors noted that the histochemical staining methods were not particularly sensitive for bacteria. Furthermore, the FISH protocol used was optimized for the detection of gram-negative organisms and did not include a pre-digestion step, which can increase the sensitivity of this technique in the detection of gram-positive or acid-fast organisms [49,50]. Given the relatively small number of cases included in this study, further studies would be required to verify the author’s conclusions.

#### 2.1.3. *E. coli* Infection within the Canine Urinary Tract

In their article published in 2014, Oliveira, Dias, and Pomba [51] briefly describe how they used FISH and PCR to study the relationship between the fluoroquinolone resistance of canine *E. coli* uropathogenic isolates and their ability to form biofilms. Biofilm production was tested using fluorescein-labelled eubacterial probes, where isolates from dogs with urinary tract infections had been cultured. It was shown that isolates with the genes for aerobactin aer, afimbrial adhesion I afa, and beta-lactamase (identified by PCR) were significantly associated with biofilm production (*p* < 0.05), and that biofilm production was significantly associated with fluoroquinolone resistance (*p* < 0.05).

The case report by Brükner [52] describes a rare instance of urinary malakoplakia in a four-month-old French Bulldog, in which *E. coli* was identified as the causal agent. The case presented with a history of recurrent urinary tract infections accompanied by pollakiuria, hematuria, and incontinence. Numerous diagnostic tests were performed including abdominal ultrasonography, serum and hematology biochemical testing, and bacterial culture; cystoscopic biopsies of the bladder wall were sent for FISH and histological examination. Clusters of invasive, intracellular, rod-like bacteria were evident below the epithelium of the affected areas of the bladder wall under FISH, consistent with *E. coli*. After 12 weeks of Enrofloxacin treatment, marked improvements were noted on follow-up ultrasonography; repeat culture and FISH of bladder-wall samples demonstrated no presence of *E. coli*. The patient was declared clear of all clinical signs—including presenting with a normal bladder under ultrasonography—59 weeks after the cessation of treatment.

#### 2.1.4. *E. coli* in Feline Inflammatory Disease 

In feline medicine, the use of FISH in the assessment of *E. coli* as the primary agent of specific disease states is limited; however, there are two case reports that have employed FISH in an investigatory or confirmatory capacity, which identified *E. coli* as a pathogen of significance within feline genitourinary and gastrointestinal disease etiology. 

Cattin, Hardcastle, and Simpson [53] reported a case of vaginal malakoplakia in a young, spayed domestic shorthaired cat, which presented with dysuria and hematuria that was unresponsive to antibiotic treatment. Further investigations of a small, fleshy erythematous mass seen protruding from the vaginal vault, revealed a sizeable irregular mass in which gram-negative bacteria were noted using periodic acid–Schiff staining; analysis using FISH (which specifically employed eubacterial, non-eubacterial, and *E. coli*-specific 16s rRNA probes) revealed the presence of invasive intracellular *E. coli*. Treatment was guided with the aid of tissue culture and antimicrobial susceptibility testing, which led to the administration of enrofloxacin over a six-week course. Complete clinical resolution of both the mass and all clinical signs was observed. This was the first observed case of invasive intracellular *E. coli* causing vaginal disease that was successfully treated with a fluoroquinolone.

In the case study of Leal et al. [54], a four-year-old cat presented with chronic intermittent hematochezia as well as fecal incontinence, which had persisted for seven months. Physical examination, blood cell counts, and biochemistry panels were within normal range, and fecal sample tests were negative for *Tritrichomonas* and *Giardia* spp. Further investigations with ultrasonography and colonoscopy revealed irregular thickening of the colonic wall; the mucosa was observed to be friable, bleeding easily during colonoscopy, with multiple erosions conducive of ulcerative colitis or infiltrative neoplasia. Biopsies were subsequently taken and submitted for histopathology and FISH. Histopathology revealed findings consistent with granulomatous ulcerative colitis, with macrophages containing PAS positive material; FISH demonstrated multifocal clusters of invasive intracellular *E. coli*. However, the authors note that more bacteria were visible when using the eubacterial probe than were visible using the *E. coli*-specific probe, indicative of possible mixed infection. Cultures produced from colonic swabs were found to be positive for *E. coli* but negative to *Salmonella* spp., *Yersinia* spp., and *Campylobacter* spp. Upon completion of antimicrobial susceptibility testing, a six-week course of enrofloxacin was prescribed, resulting in the resolution of all clinical signs, with remission of 13 months at the time of publication. 

Whilst cases of granulomatous ulcerative colitis are rare in felines, with the publication by Leal et al. [54] being the second documented case of PAS+ GUC in an adult cat, the case study published by Matsumoto et al. [55] appears to provide confirmation of the causation offered by the previous authors. In their publication, Matsumoto and colleagues noted significant numbers of phagocytosed *E. coli* within PAS+ macrophages when samples underwent histopathological and immunohistochemical examination; culture confirmed the presence of *E. coli*, sensitive to enrofloxacin, and upon receiving treatment with fluoroquinolone, signalment resolved within a 14-day period. It is important to note that FISH was not used in this case. 

### 2.2. The Use of FISH in the Detection and Visualization of Helicobacter

#### 2.2.1. *Helicobacter* within the Gastrointestinal Tract of Dogs and Cats

Historically, the pathogenicity of both gastric and enterohepatic *Helicobacter* spp. has been the subject of numerous studies in humans and animals [56,57,58,59,60,61,62]. Recordati et al. [58] applied immunohistochemistry, 16s rRNA gene sequence analysis, PCR, and FISH to assess the spatial distribution of *Helicobacter* spp. throughout the gastrointestinal (GI) tract and the role of enterohepatic variants within the GI tract of healthy dogs. These authors noted that the large intestine of dogs had sizeable colonies of *Helicobacter* spp., especially in the gastric mucosa and crypts of the cecum and colon. Specifically, they found that the stomach and large intestine were colonized by *H. bizzozeronii*, *H. salomonis*, *H. felis* (forms of gastric *Helicobacter*), and *H. bilis*/*flexispira* taxon 8, *H. cinaedi*, and *H. canis* (species of enterohepatic *Helicobacter*). It was also apparent that within the participants (all of which were healthy and showed no signs of GI disease), the small intestine carried very few *Helicobacter* spp. of either the gastric or enterohepatic form. Additionally, none were detected in the pancreas, liver, or bile. However, in the study by Polanco et al. [56] assessing the prevalence of non-*H. pylori Helicobacter* (NHPH) spp. within the gastric mucosa of 20 pet dogs, 95% of cases demonstrated mild to marked gastritis within the fundus, whether apparently healthy or otherwise. Using histopathology, PCR, and FISH, Polanco and colleagues found *H. felis*, *H. salomonis*, and uncultured *Helicobacter* spp. to be most prevalent; of the three NHPH species detected, a negative correlation was found to exist between *H. felis* and *H. salomonis* and/or *Helicobacter* spp.

Similar findings were noted by Priestnall et al. [57] when evaluating subtypes of *Helicobacter heilmannii*—of which five have been described [61]—in the gastric mucosa of both dogs and cats. The aim of their study was to determine if the forms of *H. heilmannii*- and *H. heilmannii*-like organisms (HHLOs) were the same as those in humans and to assess any zoonotic potential that may exist. Using PCR amplification and sequencing as well as FISH, the authors looked for subtypes and HHLOs, namely *H. felis*, *H. bizzozeronii*, and *H. salomonis*. They were able to ascertain that the subtypes of *H. heilmannii* in cats and dogs were predominantly types 2 and 4, which coexisted with the HHLOs *H. felis*, *H. bizzozeronii*, and *H. salomonis*. This combination differed to the predominant subtype/HHLO combination present in humans (type 1 and *H. suis*, respectively). Interestingly, it was noted that the HHLO/subtype combination differed between cats and dogs; cats also demonstrated colony variance dependent on their country of origin. In the same study, FISH failed to effectively characterize the subtypes of *H. heilmannii* with specific probes in 14 of the 15 cats included in the study, despite effective hybridization with eubacterial probes. The authors suspected that issues with the binding of specific probes may have been due to prolonged fixation with formalin prior to inclusion. 

#### 2.2.2. *Helicobacter* within the Liver and Portal System of Dogs and Cats

In the study by Greiter-Wilke et al. [63], the association between cholangiohepatitis and *Helicobacter* was investigated in cats. PCR, histopathology, immunohistochemistry, and FISH (with eubacterial and non-eubacterial 16s rRNA probes only) were used on historical FFPE samples from cats who had been diagnosed with cholangiohepatitis between 1992 and 2001 to determine the presence and type of *Helicobacter* involved. Of the 32 cases selected (lymphocytic cholangiohepatitis = 10; mixed neutrophilic and lymphocytic cholangiohepatitis = 10; lymphocytic cholangitis = 6; mixed neutrophilic and lymphocytic cholangitis = 3; lymphocytic portal hepatitis = 3; neutrophilic cholangiohepatitis = 2; and neutrophilic cholangitis = 2), only three demonstrated the presence of *Helicobacter* spp. Of the three positive samples, one was reported as being from a case of suppurative cholangitis (SC), one was from a case with lymphocytic portal hepatitis (LPH), and one was from a control case with portosystemic vascular anomalies (PSVA). Interestingly, testing with FISH only revealed a single semi-curved bacterium, described as being *Helicobacter*-like in its form, in the SC case; unfortunately, no additional tests were performed to confirm the species. This result was similar to the findings of Recordati et al. [58], where no *Helicobacter* spp. were found in the pancreas, liver, or bile of dogs. The authors state that the disparity in findings between PCR and FISH may be due to a potentially increased sensitivity of PCR or that the PCR was detecting intestinal *Helicobacter* spp. DNA, which was in circulation through the liver, rather than from in situ colonization.

#### 2.2.3. *Helicobacter* within the Enterohepatic System of Cats

When considering other conditions of the feline enterohepatic system and their association with *Helicobacter* spp., Swennes et al. [60] investigated the role of enterohepatic *Helicobacter* spp. (EHS) in the formation of non-hematopoietic intestinal carcinoma in cats. The authors included 55 cases selected using a retrospective examination of cats diagnosed with intestinal carcinoma between 1997 and 2006; 22 additional cats (euthanized) from an animal shelter without signs of GI disease were included. FFPE samples were mounted on slides, upon which histopathology was performed to classify the types of tumor present in samples from both the small intestine (SI) and large intestine (LI) (*n* = 18 and *n* = 31, respectively). The classifications used were solid carcinoma (SI = 4; LI = 1), adenocarcinoma (SI = 9; LI = 18), and mucinous adenocarcinoma (SI = 2; LI = 10); three cases with small intestinal involvement and two with large intestinal involvement were undeterminable. FISH was then performed using *Helicobacter* genus probes alongside species-specific probes, namely for *H. bilis*, *H. canis*, and *H. marmotae*. In situ hybridization revealed that EHS was prevalent in 56% of cats with intestinal carcinoma (30/54). EHS infection was present in 68% (21/31) of the LI tumor cases and in 33% (6/18) of the SI tumor cases; significantly larger numbers of *H. bilis* were noted in the LI tumor cases; and EHS infection was noted as very common (12/13, 92%) in mucinous adenocarcinoma cases. Of the latter group, increased numbers of *H. bilis*, *H. canis*, and *H. marmotae* were frequently observed. Within the control group, 15 of the 22 shelter cats (68%) were also found to have EHS present (*H. bilis* = 12/22, 55%; *H. canis* = 6/22, 27%; *H. marmotae* = 6/22, 27%) but remained free of any apparent disease afflicting the enterohepatic system. The authors conclude that EHS may play a substantial role in feline large intestinal carcinoma development due to the significance of the increased frequency of EHS in these cases. However, similar to the findings of Hoehne et al. [64], their findings do not prove a causal relationship.

## 3. Studies Using FISH to Investigate Disease States in Companion Animals, by System

### 3.1. The Use of FISH in the Identification of Intracellular Bacteria as a Cause of Chronic Gastrointestinal Disease in Cats

A number of studies which investigated the cause of chronic gastrointestinal disease in cats report multiple species of bacteria as a potential cause. Innes et al. (2007) [65] investigated bacterial association with feline inflammatory bowel disease using FISH, with an emphasis on identifying *Bifidobacterium* spp., the *Clostridium histolyticum* subgroup, *Lactobacillus-Enterococcus* subgroups, and *Desulfovibrio* spp. FISH was performed on fecal samples from 11 cats diagnosed with inflammatory bowel disease (IBD) and 34 healthy cats using probes specific to the aforementioned bacteria. Results indicated significantly higher numbers of *Bifidobacterium* spp. (*p* = 0.029, CI 95%) and *Bacteroides* spp. (*p* = 0.048, CI 95%) in healthy cats; conversely, in cats diagnosed with IBD, significantly higher numbers of *Desulfovibrio* spp. were seen, often exceeding 10^7^ in those diagnosed with IBD (10/11 (90.91%) compared to healthy cases 14/34 (41.18%). Whilst some speculation was offered surrounding the potentially protective role of *Bifidobacteria* in the development of IBD, further studies would be required to demonstrate any such relationship. 

Janeczko et al. [66] investigated the relationship between mucosal bacteria and duodenal histopathology, cytokine mRNA, and clinical disease in cats with IBD. In their study, 27 cats (17 undergoing investigation and 10 healthy controls) were assessed. FISH probes (eubacterial and organism-specific) were used to identify mucosal bacteria. It was observed that the number of *Enterobacteriaceae* present within the mucosa was significantly higher in cats with signs of gastrointestinal disease than in healthy cats. They reported that the total number of bacteria within the mucosa was strongly associated with changes within its architecture (*p* < 0.001), with an increased density of macrophages and CD3+ lymphocytes (*p* < 0.002 and *p* < 0.05, respectively). The principal abnormalities seen in mucosal architecture were villus atrophy and fusion, which were correlated with the numbers of *Enterobacteriaceae, E. coli*, and *Clostridium* spp. Notably, the upregulation of cytokine mRNA and the number of clinical signs exhibited by the affected cats were markedly increased with the presence of these bacteria. 

In a retrospective study by Nicklas et al. (2010) [67], FFPE samples from seven kittens of pre-weaning age underwent Gram staining, FISH, PCR, and ultrastructural imaging to determine features of enteroadherent bacteria, where *E. coli* had been cited as the primary cause of infection. Eubacterial probes and probes specific for *E. coli*/*Shigella* or *Enterococcus* spp. were applied alongside one another; to increase specificity, a non-Enc221 (non-*Enterococcus*) probe was added to the sample along with the inclusion of a positive and negative control in each assay. When performing their initial analysis, only two of the samples hybridized with the eubacterial and *E. coli/shigella*-specific probes. Due to the lack of hybridization with the eubacterial probe specifically, the authors included an additional pre-treatment step, which involved the application of lysozyme—a commonly used reagent that aids permeability of the cell wall of Gram-positive bacteria. After the inclusion of this step, the five remaining samples showed effective hybridization with both the eubacterial and *Enterococcus*-specific probes; the two cases that showed hybridization with the *E. coli*/*Shigella* probes only hybridized with the eubacterial probe when tested against *Enterococcus*, post lysozyme application. This allowed the conclusion that the adherent bacteria present were *Enterococcus* and, after PCR amplification, that the species involved was *E. hirae.* When considering FISH, this serves as a good example of the use of an additional pre-digestive/permeabilization step to aid in the identification of an otherwise unidentifiable cause of chronic gastrointestinal disorders. Lack of pre-digestion in previous studies, such as that by Lecoindre et al. [48], will decrease the sensitivity of this technique. 

Linton et al. [68] used FISH, culture, and conventional light microscopy with a panel of special stains in an attempt to investigate the involvement of bacteria in feline gastrointestinal eosinophilic fibroplasia (FGESF)—an inflammatory disease affecting the stomach or intestines and draining regional lymph nodes—that, until recently, had yet to be reported. The disease typically presented with sizeable, non-painful, palpably hard lesions, commonly positioned near the pylorus or ileocaecocolic junction. Of the thirteen cases recruited in their study, Linton and colleagues noted bacteria in nine, using a combination of investigatory methods as previously mentioned, but not in combination in all cases. Of these, FISH confirmed the presence of *Clostridium* in two cases, *E. coli* in one, and non-specific bacilli in three. The authors also noted discordant findings between histopathology and FISH in two additional cases; FISH failed to identify the presence of bacteria where histology did. However, the combination of FISH and histopathology did confirm the presence of a morphologically different form of bacteria in one additional case. Despite the authors using eubacterial probes (useful for the detection of most bacterial species, but not all), it is important to note that, thereafter, the study only utilized specific probes for the identification of *Clostridium* and *E. coli*. Whilst the authors also noted the presence of gram-positive bacteria, no pre-digestion stage was incorporated into their FISH protocols to ensure better adherence of probe to the target. These issues may account for the detection of organisms on histology only in the two cases where FISH failed to show the presence of any.

Investigations into the cause of neutrophilic IBD conducted by Maunder et al. [69] sought to explore the relationship between mucosally-invasive *Campylobacter* spp. And this subset of the disease. A retrospective review of duodenal biopsy samples was performed on seven cats with neutrophilic IBD and eight cats with lymphoplasmacytic IBD. FISH with 16s rRNA eubacterial probes, species-specific probes, and mRNA probes specific for neutrophil elastase were used in conjunction with histopathology and PCR. *Campylobacter coli* was seen in six of the seven cats with neutrophilic IBD compared to one of the eight cats with lymphoplasmacytic IBD. It was also noted that in the neutrophilic IBD cases, there were significant numbers of *C. coli* present in the mucosa, which were collocated closer to neutrophils than any other bacteria. The authors concluded that the identification of *C. coli* and its proximity to areas of neutrophilic inflammation indicated that *C. coli* may be able to produce neutrophil-stimulating compounds or induce the intestinal cells to produce chemicals that attract neutrophils. A small sample size and the lack of clinical data (with respect to treatment administered prior to inclusion of the cases) warrant larger, prospective studies to be conducted to corroborate the findings of this study. 

Hoehne et al. [64] used FISH to identify mucosally-invasive and intravascular bacteria in feline small intestinal lymphoma. The authors selected biopsies from 50 cats with alimentary lymphoma (33 with small cell lymphoma (SCL) and 17 with large cell lymphoma (LCL)) and 38 controls without lymphoma (normal or minimal change (NMC) on histopathology). Samples were evaluated using histopathology, immunohistochemistry, and FISH, the latter using 16s rRNA eubacterial and non-eubacterial probes with cultured samples for *E. coli*, *DH5 alpha*, and *Streptococcus bovis* in each assay. Despite possible concerns that the omission of a pre-digestive step may have hampered findings, bacteria in luminal cellular debris, the mucosa, and blood vessels were found in every case. The authors noted that bacteria were more frequent within luminal debris in SCL (*p* < 0.01) and LCL (*p* < 0.001) cases than in NMC cases, with an increased frequency within the LCL cases than in the LPE cases. Mucus within the LCL cases was also found to contain more bacteria than in the LPE cases; however, the frequency of adherent bacteria within the villus epithelium did not differ. Furthermore, mucosally-invasive bacteria were more commonly found in the LCL cases (82% vs. 18% in SCL and 3% in LPE cases), with intravascular bacteria being observed solely in those cases of LCL (29%); the serosa of LCL cases was also noted to contain increased numbers of bacteria compared to those with SCL, NMC, or LPE (57% vs. 11%, 8%, and 6%, respectively). As a retrospective study, it was unclear if the bacterial may have played a role in disease pathogenesis or if their presence was secondary to the underlying disease, leading the authors to conclude that further studies assessing the role of intramucosal bacterial in the etiopathogenesis of feline alimentary lymphoma are warranted.

### 3.2. The Use of FISH in the Identification of Intracellular Bacteria as a Cause of Chronic Gastrointestinal Disease in Dogs

Investigations into the canine microbiome using FISH, and specifically in dogs with chronic inflammatory enteropathy (CIE), have, like studies in cats, shown significant shifts in bacterial colonization dynamics. Cassmann et al. [70] demonstrated that dogs with chronic enteropathies (CE) harbored increased numbers of *Clostridium-coccoides*/*Eubacterium rectale*-group, *Bacteroides*, *Enterobacteriaceae*, and *E. coli* organisms within the ileal and colonic mucosa versus those that were disease free. In this study, histopathology was used to grade the severity of inflammation and 16s rRNA eubacterial probes with FISH to establish the total numbers of bacteria present. Subsequent analyses were performed using probes that specifically targeted individual species of bacteria. The results showed that inflammatory bowel disease in dogs was associated with increased *Enterobacteriaceae* and *E. coli* attached to epithelial surfaces or within the intestinal mucosa, with increased total numbers of bacteria being associated with increased severity of disease. Granulomatous colitis was associated solely with the presence of adherent-invasive *E. coli* within canine ileal and colonic mucosa; dogs with intestinal neoplasia were noted to have increased adherent *Enterobacteriaceae* and *E. coli*, and invasive *Enterobacteriaceae, E. coli*, and *Bacteroides*.

Giaretta et al. [71] recorded similar findings when examining the colonic bacterial biogeography of dogs, noting an increase in *E. coli*/*Shigella* spp. on the colonic surface and within the colonic crypts. However, they also found that there was a reduction in the total number of bacteria within the crypts of dogs suffering with CIE as well as a general reduction in *Helicobacter* spp. and *Akkermansia* spp. within the colon. Using 16s rRNA eubacterial and non-eubacterial probes to establish total numbers of bacteria, the authors employed probes specific to *E. coli*/*Shigella* spp., *Faecalibacterium* spp., *Akkermansia* spp., and *Helicobacter* spp. against a total of 33 canine samples—22 with CIE and 11 controls. The results led to the conclusion that not only was there an alteration in bacterial colony composition within the colonic mucosal microbiota of CIE dogs, but that cryptal bacteria appeared to be comprised primarily of *Helicobacter* spp. and this species may play a role in reducing colonization by pathogenic species or in regulating inflammation within the colon of dogs. Further large-scale studies would need to be conducted to corroborate these conclusions.

In a case of acute protein-losing ulcerative enterocolitis observed in a three-and-a-half-year-old mixed-breed dog, Cartwright et al. [72] noted multi-resistant *Enterococcus* spp. and *E. coli* as the infectious agents. Histopathology, FISH, and culture were used to elucidate a diagnosis; eubacterial and species-specific probes were applied, which revealed clusters of *Enterococcus* spp. within the mucosa of the colon but no *E. coli*. FISH showed no bacteria had invaded the ileal mucosa; however *E. coli* and a multi-resistant form of *Enterococcus faecum* were present in the cultures from both the ileum and colon. 

### 3.3. The Use of FISH in the Identification of Intracellular Bacteria as a Cause of Hepatic Disease in Cats

Warren et al. [73] reviewed 78 cases of feline chronic liver disease, which included lymphocytic cholangitis (FLC) (*n* = 51) and feline hepatic lymphoma (FHL) (*n* = 27), along with 10 control cases using standardized histopathology, B- and T-cell immunophenotyping, T-cell receptor gene rearrangement via PCR, and eubacterial FISH (both eubacterial rRNA probes and non-eubacterial probes were used with Cy3 and 6FAM fluorophores, respectively). Most notable was that 32 of 36 (88%) FLC cases submitted for FISH analysis and all FHL cases showed no presence of bacteria using eubacterial probes. As such, the authors concluded that their findings did not support the notion that bacteria have an active role in the etiology of FLC. Their findings supported those of Greiter-Wilke et al. [63], who ruled out the involvement of *Helicobacter* spp. as a causal agent of FLC.

Conversely, Twedt et al. [38] did note a relationship between the presence of bacteria and the etiopathogenesis of feline inflammatory liver disease (ILD) using immunohistochemistry, histopathology, and FISH. Concerning the latter, 16s and 23s rRNA FISH was conducted on samples from 39 cats with ILD and 19 with normal hepatic biopsies, as determined by histopathology (the results of which were classified against the World Small Animal Veterinary Association guidelines and included non-specific reactive hepatitis (*n* = 12), neutrophilic cholangitis (*n* = 12), lymphocytic cholangitis (*n* = 7), cholestasis/obstruction (*n* = 3), probable lymphoma (*n* = 3), and acute hepatitis (*n* = 2)). Using eubacterial probes and non-eubacterial probes in conjunction (Cy3 and FAM labelled, respectively), samples were initially tested for the presence of bacteria. Thereafter, simultaneous labelling was undertaken on samples using eubacterial probes and species-specific probes against *Clostridium* spp., *Bacteriodes/Prevotella* spp., *Enterobacteriaceae*, *E. coli* spp., *Helicobacter* spp., and *Streptococcus* spp. (Cy3 and 6-FAM, respectively). Findings revealed that bacteria were present in 13/31 of the IDL cases as well as 1/17 of the control cases; 8/39 of the IDL and 2/19 of the control cases demonstrated bacteria restricted to the outer liver capsule, whose presence was likely due to contamination and, as such, were excluded. Of the remaining viable IDL samples, intrahepatic bacteria were seen in or around portal vessels or venous sinusoids (*n* = 9), within the hepatic parenchyma (*n* = 3), or within the bile duct (*n* = 1); the remaining control sample was seen to have intrahepatic bacteria around the portal vessels and venous sinusoids. It was also noted that samples taken from cats with *E. coli*-positive neutrophilic cholangitis showed the highest levels of bacterial colonization. 

### 3.4. The Use of FISH in the Identification of Intracellular Bacteria as a Cause of Hepatic Disease in Dogs

In the assessment of canine hepatitis, investigations have combined FISH with other methods to identify a possible infectious etiology. Hutchins et al. [74] used FFPE liver samples to retrospectively investigate cases of canine granulomatous hepatitis using histopathology, FISH, and PCR for *Bartonella*. Of the 15 FFPE samples, 11 were subjected to FISH using eubacterial (3′ end labelled with 6-FAM) and non-eubacterial (5′ end labelled with Cy3) probes; no bacteria were seen within granulomas, bile ducts, Kupfer cells, or hepatic parenchyma despite the control samples demonstrating positive hybridization. Furthermore, histopathology performed on 13 of 15 archival samples were negative for fungi, acid-fast bacteria and Gram-negative and Gram-positive bacteria. PCR for *Bartonella* spp. was performed on all 15 FFPE samples, again returning negative results in all samples, including one from a dog with recorded *B. henselae* infection. Whilst the authors conclude that investigations using FFPE samples with the aforementioned techniques may be of low yield and that infectious agents are an unlikely cause of granulomatous hepatitis, the limitations of this study are that the length of time that samples were exposed to formalin appears to be unknown (despite the documented issues surrounding the deleterious effect of prolonged formalin exposure on the integrity of genetic material) as well as sample sizes being potentially inadequate for proper evaluation by FISH or PCR. Further studies would be warranted to confirm any conclusions made as a result of this research.

Contrary to the findings of Hutchins et al. [74], there have been cases reported where bacteria have been found in dogs with hepatic disease, where FISH has been used in conjunction with other methods to effect diagnosis. Giuliano et al. [75] noted a rare instance of acute hepatic necrosis in a young Staffordshire Bull Terrier, caused by *Salmonella enterica*. Eleven days after first developing symptoms of vomiting, loose stools, lethargy, and reduced appetite, and despite symptomatic treatment, the dog was euthanized due to the onset of hepatic encephalopathy with seizures. On postmortem examination, the liver was found to be the only organ demonstrating signs of abnormality. Samples taken for histopathology and culture returned a diagnosis of acute hepatic necrosis with the presence of monophasic *S. enterica* serotype I 4,5,12:-:1,2; Gram staining revealed the presence of Gram-negative bacteria within necrotic regions of the parenchyma. Whilst FISH was employed to confirm the presence of *S. enterica*, the authors do not elicit whether a eubacterial probe or species-specific probe was used in its detection.

McCallum et al. [76] retrospectively selected 10 dogs with suspected canine granulomatous hepatitis (GH) between 2013 and 2016 or from those that had been consulted on by telephone during the same time period. In conjunction with histopathology and PCR, FISH using 16s rRNA eubacterial probes and species-specific probes (specifically, *Campylobacter*, *Clostridium*, *E. coli*, *Helicobacter*, *Leptospira*, and *Salmonella*) was used to test for bacterial causes. On the FFPE liver samples, discreet clusters of *Leptospira* were seen in the liver of eight dogs, with two showing dispersed *Leptospira*; five cases were also identified as having multiple bacterial species present, potentially suggesting coinfection (*Campylobacter coli* + *Salmonella* spp., *n* = 1; *C. coli*, *n* = 1; *Campylobacter jejuni* + *Salmonella*, *n* = 1; *Salmonella*, *n* = 1; *Helicobacter*, *n* = 1). However, four control dogs (two with a known cause of GH and two healthy dogs) and one additional case diagnosed via autopsy as having GH alongside severe liver dysplasia were negative for *Leptospira*; the latter did show clusters of *Salmonella* spp. with some *C. coli*. The ten selected cases were treated via different pharmacological methods, based on clinician preference, with only six surviving to a median of 434 days (22–878). Three underwent further testing, with two showing the presence *L. interrogans*/*kirschneri* after FISH and PCR speciation.

Furthermore, Im et al. [77] also found intrahepatic bacteria in three cases of canine hepatitis (neutrophilic hepatitis, pyogranulomatous hepatitis, and lymphoplasmacytic hepatitis with cholangiohepatitis) using FISH, culture, histopathology, and PCR (not all used on all cases). FISH with eubacterial and non-eubacterial probes in combination provided the most consistent results of all the tests conducted. Despite a lack of sample material to be able to conduct species-specific tests, in one of the three cases, speciation was achieved through morphological findings and treatment, leading the authors to conclude that *Helicobacter* spp. was the most likely cause. In another case, speciation was achieved via the culture of bile and choledocholiths as well as through observed morphology; *E. coli* and *Enterococcus* spp. were noted as the most likely cause. In the last of the three cases, the bacteria involved were not identified, only being described as multifocal, single pleiomorphic rods occupying areas around the bile duct, hepatic vessels, and sinusoids. Despite this case series demonstrating the presence of bacteria as a potential cause of hepatitis (a hypothesis reinforced by treatment with antibiotics and subsequent recovery), it included no control cases for comparison and, thus, no definitive conclusion could be made as to bacterial involvement in these cases of hepatitis. This was also an issue noted by McCallum et al. [76], whose study could not definitively establish the role of *Leptospira* spp. in GH cases, not only because of the persistent/recurrent infection regardless of treatment against that organism, but also because of a lack of a disease-free control group with which to compare. Whilst there appears to be a growing evidence base for the involvement of infectious agents in both canine and feline hepatitis, further large-scale, adequately-powered studies would be required to make any definitive conclusions as to the role of bacteria in this disease.

### 3.5. The Use of FISH in the Identification of Intracellular Bacteria as a Cause of Urinary Tract Infections in Dogs

Surprisingly, FISH has not been used to investigate *Leptospira* spp. as a potential cause of kidney disease in either cats or dogs, despite numerous studies using other methods demonstrating their presence within the renal systems of both [78,79]. However, the study of Hutton et al. [80] aimed to use FISH alongside histopathology—specifically, using modified Steiner (MS) and immunohistochemistry (IHC)—and PCR to investigate the presence of *Borrelia burgdorferi* as a potential cause of Lyme nephritis in dogs. Of the 36 samples retrospectively selected from Lyme-positive dogs diagnosed between 1996 and 2004 (26 affected and 10 controls), 2 showed positive results using FISH. Both PCR and IHC showed no positive results for *B. burgdorferi*, with MS staining returning one positive case within the control group. Nevertheless, bacteria—not of the species *Borrelia*—were detected using eubacterial primers and probes for PCR and FISH (PCR *n* = 22/26, 85%; FISH *n* = 10/21, 48%) in positive cases. FISH also revealed that 50% of the control cases had eubacterial organisms present within the kidneys; in an additional five cases (four positive cases and one control), bacteria were detected on the external surfaces of the biopsies, presumed to be contamination and not as a result of infection. Furthermore, the authors noted that in 20/31 cases (66% of all cases; 14/21 (66%) of affected cases and 6/10 (60%) of control cases) upon which both FISH and PCR were performed, discrepancies in the results existed between the two testing methods. Again, this was cited as most likely due to contamination, as samples were unlikely to have been handled using aseptic techniques when originally collected. The authors concluded that due the lack of *B. burgdorferi* and the consistent lack of coinfecting agents, Lyme nephritis was most likely caused as a result of postinfectious glomerulonephritis secondary to Lyme antigen-associated immune complex deposition within the glomeruli. Further research would be required to confirm this hypothesis.

Other areas of the urinary tract have also been assessed for the presence of bacteria as a source of localized disease. Borys et al. [81] investigated the etiology of proliferative urethritis (PU) and its association with bacterial cystitis in dogs. This retrospective study recruited 27 dogs (22 symptomatic cases and 5 euthanized cases without urinary tract involvement) whose records were suggestive of PU or granulomatous urethritis and where samples had been taken between 1986 and 2016. Samples were subjected to culture (both urine and tissue culture) and histopathology. Where sufficient urethral tissue samples remained (of which there were 13), FISH was conducted using 16s rRNA eubacterial probes in conjunction with non-eubacterial probes as a negative control. Upon the visualization of bacteria, subsequent analysis was conducted using species-specific probes for *E. coli*, *Streptococcus* spp., and/or *Staphylococcus* spp. depending on morphology. Of these 13 samples, 7 (54%) returned positive results for bacteria, the majority of which were adherent and invasive in nature (5/13, 39% vs. 2/13, 15% adherent only). Furthermore, 11/13 samples underwent FISH analysis where it was noted that perioperative aerobic bacterial urine culture (ABUC) had been conducted previously; 7/11 were ABUC negative, of which 3 returned positive results for bacteria using FISH. In addition, a total of five dogs returned positive FISH results where culture results (ABUC, tissue culture or both) were negative. Rods were visualized in the urothelial tissue of six dogs; two stained positive for *E. coli* (both positive for the same organism via ABUC); one other was positive for *Enterococcus* within the bladder mucosa (ABUC negative); and another was positive for *Staphylococcus*, present within urethral plaques and clustered within the epithelium of the bladder (*S. psuedintermedius* was noted within mixed bacterial growth under ABUC). FISH also revealed adherent-invasive bacteria in four samples that returned positive results via staining for species-specific pathogens. 

### 3.6. The Use of FISH in the Identification of Intracellular Bacteria as a Cause of Integumentary Infections in Dogs

Whilst the previous study noted a case were *Staphylococcus* was present within the lower urinary tract, this group of bacteria are also commonly associated within skin conditions in companion animals [82]. In the prospective study by Ravens et al. [83], both FISH and histology were used to confirm that bacteria were adequately retrieved from cases of canine superficial bacterial pyoderma (SBP). In total, 27 dogs with SBP were selected to compare three sampling methods—a dry cotton swab, a cotton swab moistened with saline, and skin scraping—and their ability to provide adequate yield for bacterial culture. FISH, employing 16s rRNA eubacterial and non-eubacterial probes on FFPE sections of biopsied skin, confirmed the location of bacteria as discovered using histopathology (specifically, via hematoxylin-eosin, Giemsa, and Gram–Twort staining) in all samples. Histopathology revealed organisms within the non-follicular epithelium, in the hair follicle ostia and lumina, as well as clustered within the stratum corneum associated with neutrophilic crusting; FISH confirmed these findings. Of the samples collected, *S. pseudintermedius* was present in 24/27 dogs and *S. schleiferi* was present in 3/27. 

Continuing with the dermatological and confirmatory applications of FISH, the study of Banovic, Linder, and Olivry [84] also detected *Staphylococcus* spp. within the intracorneal region in cases of canine exfoliative superficial pyoderma with epidermal collarettes. In their pursuit of characterizing the clinical, cytological, microbial, and histopathological features of the collarettes, five dogs were selected for inclusion based on the presence of at least one actively expanding collarette with an erythematous rim that had developed over the preceding 48 h. Samples underwent cytology, culture, and susceptibility testing as well as *Staphylococcus* spp. genotypic relatedness testing using pulsed-field gel electrophoresis with SmaI enzymatic digestion. PCR was then used to determine the presence of *S. pseudintermedius* exfoliative toxin genes within the isolated strains. Finally, histopathology, FISH, and immunofluorescence were performed on FFPE biopsy samples; immunofluorescence was used for immunomapping, HE staining, and FISH (with 16s rRNA eubacterial and species-specific probes) were used to confirm the location of bacteria. The authors noted that *Staphylococci* were present in the leading cleft and that Gram-positive cocci were predominantly present in spongiotic foci and focal crusts. They concluded that epidermal collarettes should be considered unique clinical and histological lesions of exfoliative superficial pyoderma, distinct from those of impetigo and superficial bacterial folliculitis.

### 3.7. The Use of FISH in the Identification of Intracellular Bacteria as a Cause of Cardiovascular Disease in Dogs

As previously mentioned, *Staphylococcus* spp. are predominantly associated with skin disorders but have been found to cause disease affecting other regions [85]. One such addition of note is the study of Kornreich et al. [86], who used FISH to identify the causative species of bacteria responsible for canine bacterial endocarditis (BE). Using archival FFPE valve sections from 17 dogs retrospectively selected from cases between 1990 and 2007, both histopathology (using Gram, HE, and modified Steiner’s stains) and FISH (using 16s rRNA eubacterial, non-eubacterial, and species-specific probes) were performed to elucidate the bacterial species most commonly responsible. Overall, 7/17 samples returned a positive result for bacteria using FISH and eubacterial probes; histology returned positive results for bacteria in 4 cases using HE stain, 6 cases using Gram stain, and 8 using modified Steiner’s stain. Of the seven positive FISH samples, additional analysis with species-specific probes was conducted, with probes being selected against bacterial morphology and Gram staining. Four cases returned positive for *Staphylococcus* spp., one for *Streptococcus* spp., and three for both *Staphylococcus* and *Streptococcus*; none were positive for *Bartonella* spp. 

### 3.8. The Use of FISH in the Identification of Intracellular Organisms as a Cause of Cardiovascular Disease in Cats

Donovan et al. [87] investigated potential causes of feline endomyocarditis-left ventricular endocardial fibrosis complex (FEMC–LVEF) and found that *Bartonella* spp. may play a primary role. The authors used PCR, DNA sequencing, histopathology, and FISH on FFPE samples from 60 cats (FEMC–LVEF, *n* = 36; hypertrophic cardiomyopathy (HCM), *n* = 12; no cardiac disease, *n* = 12) retrospectively selected from cases between 1999 and 2014, as listed within the digital pathology database of the Animal Medical Center, New York. PCR identified *Bartonella* spp. in the hearts of 18/36 FEMC–LVEF cats; of these, 10 were positive for one form *Bartonella* spp. and 8 were positive for multiple species. Of the HCM samples, only one was positive for *Bartonella*, specifically *B. koehlerae*. FISH was performed on seven of the PCR-positive FEMC–LVEF samples using eubacterial and non-eubacterial probes initially, which returned four positive cases. Hybridization was subsequently conducted on those samples using *Bartonella* genus-specific probes, for which all samples tested positive. Further sequencing demonstrated the presence of *B. vinsonii* subsp. *berkhoffii* in two of the four FISH-positive samples (one genotype I and the other genotype III)—*B. vinsonii* subsp. *berkhoffii* genotype I and *B. clarridgeiae* in one, and *B. henselae* San Antonio 2 in another. The authors noted some limitations, namely that confirmation of coinfection via PCR was difficult due to preferential amplification of organisms with the highest number, which may have affected their results; sample age was significant in some cases and thus degradation of DNA may have occurred; other causes of cardiac inflammation were not systematically tested; special stains or IHC were not performed on all cases; and FISH was not able to localize bacteria in three cases, possibly because of non-homogeneous *Bartonella* spp. myocardial distribution or due to excessive fixation times. Thus, direct causality between FEMC–LVEF and the presence of *Bartonella* spp. could not be established. Further studies, prospective in nature, would be required to demonstrate any association.

Within the circulatory system of cats, *Mycoplasma* spp. have also been detected on the surface of red blood cells (RBS), as reported in the study of Peters et al. [88]. Their study utilized ten cats, experimentally infected with *Mycoplasma haemofelis*, “*Candidatus Mycoplasma haemominutum*”, or “*Candidatus Mycoplasma turicensis*”, to identify which tissue types facilitated their persistence and survival; this was facilitated using FISH with species-specific 16s rDNA probes, qPCR, and histopathology. Samples were taken post-mortem at various time points to ensure collection during the acute phase of infection (to determine cell types most prone to infection) as well as during the chronic phase (to determine cell types and organs that may be involved in sequestration and copy-number cycling). Tissue samples were taken from the tonsils, submandibular salivary glands, bone marrow, lungs, liver, spleen and kidney, jejunum, colon, mesenteric lymph nodes, and colonic lymph nodes. Results showed that within these tissues, *M. haemofelis* was localized to the erythrocytes during both phases of infection, with no indication of their presence within other cell types. These findings supported those of an earlier study by Berent et al. [89] in which *M. haemofelis* was found only in the erythrocytes of feline liver and kidney samples that underwent testing with non-fluorescent in situ hybridization. However, Peters et al. [88] noted that in their study, the sensitivity of FISH was limited and could not rule out the possibility of additional sites of disease persistence/sequestration. This limitation was attributed to a potential lack of organism numbers, the potential for loss of bacteria during sample processing for FISH, and a potential lack of ability for the probe to bind to its target. Furthermore, the need for tyramide amplification of probe output, used to overcome erythrocyte autofluorescence [90,91,92], was required to elicit a signal; without this step, no probe emission was detectable. 

## 4. Conclusions

To date, there are over 70 variations of FISH reported within the literature with techniques using RNA, DNA, PNA and LNA probes. FISH can be performed on a variety of different sample types from living samples, fresh tissue, frozen tissue, and formalin-fixed tissue. Tests can be run individually or multiplexed to combine probes against multiple samples, simultaneously [18,19,21,34,93,94,95,96,97,98,99,100,101,102,103].

FISH has proven to have diverse applications in many different fields of medicine and research. Within the realms of small animal medicine, especially where FISH has been used to identify bacterial causes of disease, its application has been predominantly in the investigation of gastrointestinal and hepatic disease. There is a relative paucity of research looking at the potential role that intracellular bacteria may have in disease states affecting other organ systems, especially within feline medicine. The sentinel publication in veterinary medicine, which increased awareness of FISH techniques, was the discovery of *E. coli* as the cause of canine granulomatous colitis [43]. The protocol reported the use of Eubacterial probes to initially screen for bacteria, and later included further identification with *E. coli*-specific probes. Since this time, multiple studies have screened for bacterial etiologies in a range of diseases affecting the gastrointestinal, urinary, or integumentary system. Whilst FISH has proved useful in many of these studies, negative results should be interpreted with caution as the sensitivity of the technique is dependent upon the optimization of the protocol to the organism and tissue of interest [11,12,50,104].

Currently, for many diseases where an underlying causative agent is unknown, FISH is best employed alongside IHC, PCR, and/or culture as part of a screening protocol. However, as our knowledge of different disease etiologies expands and techniques are optimized to the tissues and organisms of interest, FISH could serve as a tool whose primary use is not just to detect the presence of organisms but more for the provision of information on the spatial distribution, viability, and number of organisms present within a given sample. This would allow for a greater understanding of how and where organisms populate a host, how multiple organisms coexist within that host, and how these factors relate to disease state and disease progression.

## Data Availability

All data are contained within the article.

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
