# Peer review of "Fluorescent In Situ Hybridization for the Detection of Intracellular Bacteria in Companion Animals"

_vetsci, 2024, doi:10.3390/vetsci11010052_

Round 1

Reviewer 1 Report

Comments and Suggestions for Authors

This review titled ‘’Fluorescent in situ hybridization for the detection of intracellular bacteria in companion animals’’ by Rolph et al.,  provides a good overview and significance of FISH in veterinary medicine. The application of FISH techniques for visualisation and identification of intracellular bacteria in companion animals are discussed.

Please discuss on below mentioned points:

FISH proved more sensitive than traditional histology in identifying bacteria, likely due to its ability to detect non-viable organisms.

A brief overview of various steps of FISH protocol can be discussed.

Key parameters to be considered during the design of the FISH probes can be included.

Advantages of fish over other methods of detection can be discussed.

Specific examples can be included in the introduction while discussing various limitations of the FISH.

Line 357-358 : FISH failed to identify the presence of bacteria where histology did.

The possible reasons for the failure should be discussed.

Line 267-269 : since FISH probes are specific to DNA and RNA that might occupy only certain regions of the cell, deducing the shape of the bacteria using this techniques might not be accurate.

Please Italicize the organism names

Author Response

Thank you for your time in reviewing this manuscript.  Please find our responses to your very helpful comments attached.

Reviewer 2 Report

Comments and Suggestions for Authors

I thank the authors for their work of data collection and analyses. In the discussion and conclusions, I think they could provide more precise indications on the use of this diagnostic technique.

Comments on the Quality of English Language

Minor editing of English language required.

Author Response

(The authors gave the same response as above.)

Round 2

Reviewer 2 Report

Comments and Suggestions for Authors

I thank the Authors for their work. I appreciated how they ammended the manuscript.